

# Emergent fractal phase in energy stratified random models

**Anton G. Kutlin[1,2]⋆ and Ivan M. Khaymovich[1,2]**

**1** Max-Planck-Institut für Physik komplexer Systeme,
Nöthnitzer Straße 38, 01187-Dresden, Germany
**2** Institute for Physics of Microstructures, Russian Academy of Sciences,
603950 Nizhny Novgorod, GSP-105, Russia

⋆ anton.kutlin@gmail.com

## Abstract

We study the effects of partial correlations in kinetic hopping terms of long-range disordered random matrix models on their localization properties. We consider a set of models interpolating between fully-localized Richardson's model and the celebrated Rosenzweig-Porter model (with implemented translation-invariant symmetry). In order to do this, we propose the energy-stratified spectral structure of the hopping term allowing one to decrease the range of correlations gradually. We show both analytically and numerically that any deviation from the completely correlated case leads to the emergent non-ergodic delocalization in the system unlike the predictions of localization of cooperative shielding. In order to describe the models with correlated kinetic terms, we develop the generalization of the Dyson Brownian motion and cavity approaches basing on stochastic matrix process with independent rank-one matrix increments and examine its applicability to the above set of models.



# 1  Introduction

Fractality and multifractality are intriguing and rich phenomena quite widely spread in real world, including coastline profiles, turbulence or even heartbeat and financial dynamics. In quantum systems, multifractal wave functions sharing the properties of metallic (ergodic) and insulating phases open the gap for the ergodicity breaking without a complete localization. First found at the Anderson localization transition [1,2] in single-particle disordered systems, such non-ergodic extended states have been recently claimed to describe wave-function properties of a whole many-body localized (MBL) phase in the Hilbert space of strongly interacting disordered quantum systems [3–8]. The complexity of the above systems makes the numerical simulations of them to be difficult. Indeed, the description of the MBL transition is a highly demanding problem which shows controversial numerical results not only on the critical disorder value [9–11], but on the existence of the MBL phase itself [11–13].

Therefore it is of particular concern and high demand to model essential attributes of such systems in less complicated single-particle or random matrix settings. Here, in order to simulate the multifractal eigenstate structure of the MBL phase one has to realize the random matrix models with the *robust* multifractal phase(s) in them.

However, the only random-matrix platform known so far to show robust fractal [14–21] or multifractal [22–24] properties is the family of the so-called Rosenzweig-Porter (RP) ensembles [25] and some Floquet-driven systems [26–28] showing the same effective Floquet Hamiltonian. All these models are inevitably long-range and given by the complete graphs with different statistical properties of on-site (diagonal) disorder and matrix hopping terms. Recently the interest to such long-range models and robust multifractality in them has been stirred up by their relevance and an immediate applicability of their multifractal paradigm to the description of black-hole physics based on the Sachdev-Ye-Kitaev model with diagonal disorder [8,29] and to the speeding up the algorithms of quantum annealing [30–32] and teaching of weak learners in artificial neural networks [33].

Nevertheless, the disordered long-range models undergoing Anderson localization transition are not just purely idealized mathematical objects living in the brains of scientists. By contrast, being suggested in 1980s [2, 34–36], they have been realized in the variety of physical systems spreading from trapped ions [37,38], ultra cold polar molecules [39], magnetic [40,41], and Rydberg [42,43] atoms to nitrogen vacancies in diamond [44], photonic crystals [45], nuclear spins [46], and Frenkel excitations [47]. All these realizations confirm that the presence of the long-range hopping matrix elements indeed may delocalize the system and restore the localization transition even in low-dimensional systems where the short-range models are known to be completely localized [48].

In addition, the disordered many-body systems considered in the Hilbert space [6] and their counterparts on the hierarchical graph structures [22, 23, 49, 50] have been recently

mapped (within some approximations) to the above Rosenzweig-Porter-like models. In these mappings the all-to-all hops which amplitudes depend on the Hilbert space dimension $N$ emerge naturally from the short-range models in the hierarchical (Hilbert) space. Indeed, the all-to-all coupling is obtained in these models from the consideration of long series of quantum transitions to (at least) exponentially growing number of available configurations at the certain hopping distance, while the size-dependence of the hopping amplitudes is related to the exponentially decaying propagators [6]. Proliferating this process up to the Hilbert space diameter, one immediately restores the complete graph with the $N$-dependent amplitudes [49].

However, with all the above mentioned success in applications [6, 8, 22, 23, 29–33, 49], the RP model has some caveats. Indeed, it is formed by a complete graph with the *independent identically distributed* (i.i.d.) hops, while in the models mapped to the short-range many-body disordered models the hopping terms in the Hilbert space are strongly and non-trivially correlated due to the dominance of far-away resonances in the corresponding long series of quantum transitions [51].

Taking account of such correlations brings another surprise into play. Indeed, in the case of *completely* correlated hopping amplitudes (see, e.g., disordered Richardson's model [56, 57] and Burin-Maksimov model [58, 59]) even the long-range character of hopping terms cannot delocalize the majority of the states. These effects called in the literature the cooperative shielding [60, 61] and the correlation-induced localization [21, 62–66] are based on the observation that in the disorder-free versions of such systems due to the correlated nature of the kinetic long-range terms there is measure zero of high-energy states that take the most spectral weight of the hopping and effectively screen the bulk states from the off-diagonal matrix elements. The coexistence of few high-energy states with the nearly degenerate bulk states forms a kind of energy stratification, when measure zero of states are separated from each other and from the rest of the spectrum. In the disordered setting, the above screening effectively returns the majority of the eigenstates to the short-range disordered situation with the localization described by the previously known results [48]. The high energy of a few stratified spectral edge states prevent them from being localized.

Here, a question immediately arises: to what extent such a correlation-induced localization is fragile to the reduction of the correlations? On one hand, the answer to this question has been partially given in [62, 63, 66, 67]: (i) the localization properties of the systems with the correlations respecting only the translation-invariant (TI) symmetry are nearly indistinguishable from the ones in the uncorrelated systems [63]; (ii) the addition of a tiny fraction of delocalized *uncorrelated* hopping to completely correlated models breaks down the localization in them [67], while (iii) the correlation-induced localization survives under the addition of *finite* staggered potential [62] or anisotropy-induced quasi-disorder [66] terms. But on the other hand, the partial correlations (not just mixture of fully-correlated and fully-uncorrelated terms) were out of scope of these papers.

Therefore in this paper we would like to combine the above mentioned energy stratification of measure zero of levels in the disorder-free case, common for both the correlation-induced localization and cooperative shielding, with the *partial* correlations in the hopping terms. As a remarkable example we consider the full set of models interpolating between the fully correlated Richardson's model (RM) [52–54][1] and the Rosenzweig-Porter with translation-invariant correlations (TIRP) [63].

Our choice of the TIRP model which has the same wave-function statistics as the conventional uncorrelated RP model is motivated by the simple (plain wave) eigenbasis of the kinetic term allowing one straightforwardly interpolate between the above models keeping this hopping term basis fixed. On the other hand, although the wave-function phase diagram of TIRP

---

[1]Here we focus on the localization properties in a single-particle sector of the Richardson's model [56,57]. For further information on the localization properties in the many-body sectors, one can consult [55].

is the same as the one of RP, there is no rigorous method to describe this model. Indeed, both the Dyson Brownian motion (DBM) [68] and the cavity method [69] working for fully uncorrelated models like RP [15, 24] are not applicable to the set of models of our interests. Therefore in this paper we develop another method based on the spectral decomposition of the kinetic term and the Sherman-Morrison formula for the Green's functions, generalizing the above methods, and benchmark this powerful tool with respect to the numerical results.

## 2 Model

We consider an ensemble of random $N \times N$ Hamiltonians $H = H_0 + V$ with the on-site disorder given by the matrix $H_0$, diagonal in the coordinate basis of sites $|i\rangle$, having i.i.d. random entries $\varepsilon_i$ with zero mean and unit variance on the diagonal and perturbed by a hopping matrix $V$. This hopping matrix $V$ is assumed to be infinitely long-range with a certain eigenbasis $\{|g_k\rangle\}$. Further for simplicity we consider $\{|g_k\rangle\}$ to be plain waves[2] with an integer wave number $0 \le g_k < N$ to the state's index $k$ correspondence chosen randomly for each realization, i.e.

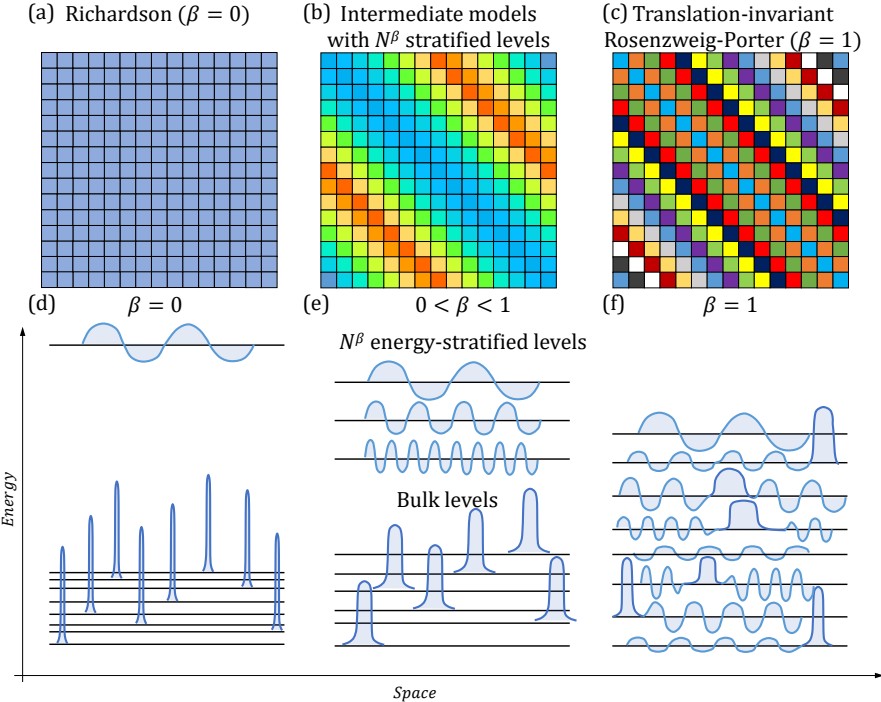

Figure 1: Graphical representation of a hopping matrix (a-c) and the correspondent spectra with $R = N^\beta$ energy-stratified levels (d-f) for (a,d) the Richardson's model (RM), $\beta = 0$, (b,e) intermediate models with $0 < \beta < 1$, and (c,f) the translation-invariant Rosenzweig-Porter model (TIRP), $\beta = 1$. The similarity in colors in (a-c) represents the degree of correlations of the hopping term $V$ of the models: from fully-correlated RM (the same color for all) to translation-invariant uncorrelated TIRP (all diagonals of different colors). The widths of the blue peaks in (d-f) represents the degree of delocalization of the wave functions on the corresponding levels (black horizontal lines).

---

[2]All the results are not sensitive to this choice: the only important assumption is that the coefficients $\langle i|g_k\rangle \sim 1/N^{1/2}$ are of the same order and, thus, ergodic in the coordinate basis.

$$H_0 = \sum_{i=1}^{N} \varepsilon_i |i\rangle\langle i|, \quad \langle \varepsilon_i \rangle = 0, \quad \langle \varepsilon_i \varepsilon_j \rangle = \delta_{ij}, \tag{1a}$$

$$V = \sum_{k=1}^{R} v_k |g_k\rangle\langle g_k|, \quad \langle i|g_k \rangle = \frac{1}{\sqrt{N}} e^{\frac{2\pi i}{N} g_k i}. \tag{1b}$$

In order to describe the RM and TIRP (as well as the whole class of models interpolating between them) we consider the hopping matrix $V$ to have the rank $R = N^\beta$, i.e., the number of non-zero real eigenvalues $v_k \neq 0$, see Figs. 1(d-f). Then RM is characterized by the rank $R = 1$ matrix $V$ ($\beta = 0$) when all the hopping elements are completely correlated to each other (see Fig. 1(a)): e.g., for $g_1 = 0$, i.e. $\langle i|g_1 \rangle = N^{-1/2} = const$

$$V^{RM} = v_1 |g_1\rangle\langle g_1| \quad \Longleftrightarrow \quad V_{ij}^{RM} = \frac{v_1}{N}, \quad \left\langle V_{ij}^{RM} V_{mn}^{RM*} \right\rangle = \frac{\langle v_1^2 \rangle}{N^2} > 0, \tag{2}$$

while TIRP model corresponds to $R = N$ ($\beta = 1$) with $N$ i.i.d. Gaussian random numbers $v_k$ providing as well $N$ i.i.d. Gaussian random TI-coefficients $V_{ij}^{TIRP} = V_{i-j}$, Fig. 1(c)

$$\left\langle V_{i-j} \right\rangle = 0, \quad \left\langle V_{i-j} V_{m-n}^* \right\rangle = \delta_{i-j,m-n} \frac{\langle v_k^2 \rangle}{N}. \tag{3}$$

The phase diagram of both above models is determined by the scaling of $V_{i,j} \sim N^{-\gamma/2}$ with a certain parameter $\gamma$ taken from the consideration of RP model [14]. The standard locator expansion converges at $V_{i,j} \sim N^{-1}$, $\gamma = 2$, providing the localization of all eigenstates at $\gamma > 2$. However the difference in correlations (2) and (3) makes the phase diagrams of RM and TIRP completely different at $\gamma < 2$, see the top and bottom lines in Fig. 5.

Indeed, like in the RP model, its translation-invariant counterpart, TIRP, shows ergodic and fractal wave-function coefficients in the coordinate basis for $\gamma < 1$ and $1 < \gamma < 2$, respectively. Unlike this, in RM $N-1$ eigenstates are (critically) localized for all $\gamma < 2$ [57]. The only level with the large eigenenergy $E \sim v_1 \sim N^{1-\gamma/2}$ is given by a slightly perturbed plain wave $|g_1\rangle$ [57,63] and provides the energy stratification, Fig. 1(d).

In the above parametrization when $V_{ij} \sim N^{-\gamma/2}$ for a general value of $R = N^\beta$ the i.i.d. Gaussian random variables $v_k$ have zero mean and the fixed variance given by

$$\langle v_i \rangle = 0, \quad \left\langle v_i v_j \right\rangle = \delta_{ij} N^{2-\gamma-\beta}. \tag{4}$$

Therefore, like in RM with $\beta = 0$, Fig. 1(d), in a more general case with $0 < \beta < 2-\gamma$ one realizes the energy stratification in the hopping term $V$ where a zero fraction $R/N \sim N^{-(1-\beta)}$ of states has an enhanced energies $v_k \sim N^{1-(\gamma+\beta)/2}$, Fig. 1(e), while the rest of the states are degenerate in the disorder-free setting $H_0 = 0$.

The presence of $R \ll N$ non-zero $v_k$ provides the correlation between TI-hopping terms $V_{i-i'} \equiv V_{\Delta i}$, Fig. 1(b). Indeed,

$$\left\langle V_{\Delta i} V_{\Delta j}^* \right\rangle_{\{v_k\}} = f(\Delta i - \Delta j) N^{-\gamma}, \tag{5}$$

where $\langle \ldots \rangle_{\{v_k\}}$ stands for the average over the hopping eigenvalues $\{v_k\}$ and the function $f(x)$ for each fixed choice of $\{g_k\} \longleftrightarrow \{k\}$ correspondence is given by the sum

$$f(x) = \sum_{k=1}^{R} \frac{e^{\frac{2\pi i}{N} g_k x}}{R}, \tag{6}$$

jumping from $f(0) = 1$ to $f(x \neq 0) \sim R^{-1/2}$. Of course for different sets of $\{g_k\}$ the correlations (5) given by $f(x \neq 0)$ have different signs.[3]

A set of such Hamiltonians with different values of $\beta$ smoothly connects RM ($\beta = 0$) with the TIRP model ($\beta = 1$), i.e. interpolates between a maximally structured translation-invariant model to a TI model with a maximally unstructured hopping while preserving an amplitude of the off-diagonal entries. This is of course not the only way to go between these two models but definitely the interesting one since by a similar interpolation one can in principle, rank by rank, reach any model, not only TIRP one.

In this work, we focus on the calculation of the fractal dimension $D(\beta)$ defined via the inverse participation ratio of the eigenstates $|\psi_E\rangle$ in the spectral bulk for the model (1) with $0 < \beta < 1$ as[4]

$$\text{IPR} = \sum_i \langle i|\psi_E\rangle^4 \sim N^{-D(\beta)} \,. \tag{8}$$

From the previous studies (see, e.g., [63]) the fractal dimensions for the limiting cases of $\beta = 0$ and $\beta = 1$ are known. Indeed, as for $\beta = 0$ and $\gamma < 2$ the eigenstates are critically localized, one should have $D(0) = 0$, while for $\beta = 1$ and $\gamma < 2$,

$$D(1) = \min(1, 2 - \gamma) \,. \tag{9}$$

In the intermediate regime, $0 < \beta < 1$, following the paradigm of the cooperative shielding [60] one expects to have zero fractal dimension $D(\beta) = 0$ as soon as there are energy stratified eigenstates with the extensive level spacing

$$\delta_k \simeq \frac{\sqrt{\langle v^2 \rangle}}{R} = N^{1-\gamma/2-3\beta/2} \gg 1 \,, \tag{10}$$

i.e., at least for $\beta < (2-\gamma)/3$. These $N^\beta$ levels are detached from the bulk of the spectrum, while the typical states in the spectral bulk are shielded from the long-ranged hopping by these high-energy modes and stay virtually unchanged after an orthogonalization to the detached ones.

However, surprisingly, this guess fails. Indeed, as we show further analytically and numerically, Fig. 2, the fractal dimension is given by the simple formula

$$D(\beta) = \min(\beta, D(1)) \,, \tag{11}$$

and shows the immediate *emergence of eigenstate fractality* of mid-spectrum states as soon as the number $N^\beta$ of energy stratified states becomes extensive, $\beta > 0$ (unlike the case of the Burin-Maksimov model [62–64]). This is the main result of our paper. Such a fractality emergence shows the fragile nature of the correlation-induced localization to partial reduction of correlations like in the case of mixture [67] of completely correlated and completely uncorrelated hopping terms.

Before going to the technical calculations, let us give qualitative arguments in favor of the above result. As one can see from the Fig. 2, the model has two regimes which differ by

---

[3]Formally, on average over the sets of $\{g_k\}$, $1 \leq k \leq R \ll N$, it gives

$$\left\langle V_{\Delta i} V_{\Delta j}^* \right\rangle = \begin{cases} N^{-\gamma}, & \Delta i = \Delta j \\ 0, & \Delta i \neq \Delta j \end{cases} \tag{7}$$

as in the TIRP model. Moreover, this works at any $\beta$ including the Richardson's model ($\beta = 0$) due to the randomness of the basis vectors $|g_k\rangle$. However, as we will see below the presence of the higher-order correlations like $\left\langle V_{\Delta i} V_{\Delta j} V_{\Delta k}^* V_{\Delta i+\Delta j-\Delta k}^* \right\rangle$ interpolates the phase diagram between TIRP and Richardson's models. The washing out of the two-point correlations in this case are not important for the further consideration.

[4]Since TIRP doesn't possesses multifractality [63], we do not consider $D_q$ for different $q$. Another definition of this quantity which we actually use will be given via self-energy in Sec. 5.4.

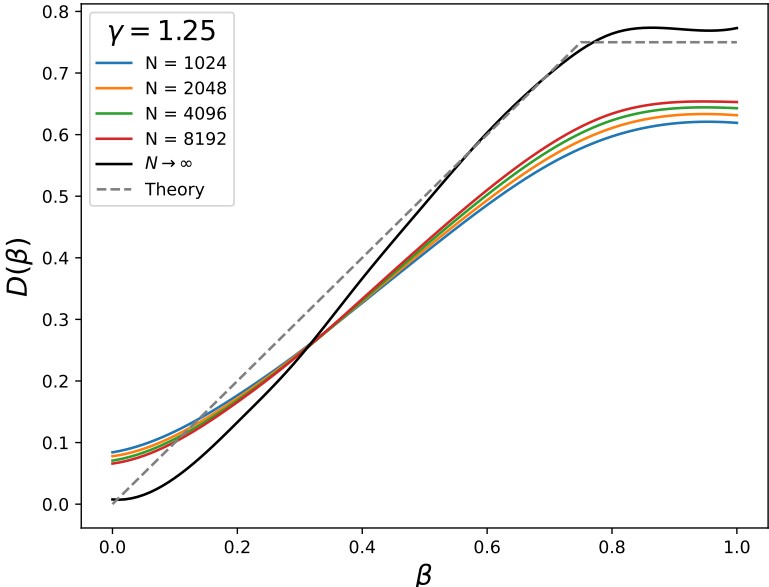

Figure 2: A fractal dimension dependence on hopping matrix rank $R = N^{\beta}$ for the model (1) with scaling parameter $\gamma = 1.25$. The data was averaged over 1000 samples for each size and extrapolated to $N \to \infty$.

the ratio of the bandwidths of diagonal $\sigma(H_0) \sim N^0$ and hopping $\sigma(V) \sim N^{1-\gamma/2-\beta/2}$ matrix terms.

1. In the case of $\sigma(H_0) \gg \sigma(V)$, $\beta > 2 - \gamma$, there is no energy-stratified states and the matrix $V$ can be considered as a perturbation in the sense of the Fermi's golden rule. The corresponding fractal dimension can be estimated similarly to the usual RP case as $D = 2 - \gamma$. Though, one cannot take this estimate on faith without a reservation in a generic model as the hopping matrix $V$ is correlated.

2. Unlike this, as soon as $\sigma(H_0) \ll \sigma(V)$, $\beta < 2 - \gamma$, almost all $N^{\beta}$ finite-energy eigenvectors of $V$ have extensive energies $v_k \gg N^0$ and form a high-energy subspace $S = span(\{|g_k\rangle\})$ of states which cannot be significantly hybridized by the presence of a "small" matrix elements of $H_0$. However, this $H_0$ matrix does hybridize the rest $N - N^{\beta}$ degenerate eigenstates of $V$ and push them forward towards the localization. Though, since these hybridized eigenstates should be orthogonal to the high-energy subspace $S$ and also orthonormal between each other, their typical support set is of the order of $dim(S) = N^{\beta}$ giving the desired linear dependence $D = \beta$.[5]

In order to derive Eq. (11) mathematically, one has to generalize the known methods (such as the cavity method of the Dyson Brownian motion) to the case of the correlated hopping terms (5). This we will do in the next sections.

---

[5]This results is easy to understand by counting the degrees of freedom and orthogonality conditions. Each of $N - N^{\beta}$ states with the fractal dimension $D$ should have enough number $2 \cdot N^D$ of degrees of freedom to be orthogonal to the rest and to each other. The unitary matrix of the size $N^D \times N^D$ formed by those states, which live on intersecting supports, has $\left(N^D\right)^2$ real degrees of freedom and guaranties their orthogonality to each other. The orthogonality to other $N - N^{\beta} - N^D$ non-stratified states is satisfied due to the support mismatch. At the same time, the number of real orthogonality conditions to $N^{\beta}$ stratified states is $2 \cdot N^D N^{\beta}$. Thus, we have $D \geq \beta$ and our model just saturates this bound.

## 3 Local resolvent as observable of interest

Before going to a newly developed method, we would like to describe an observable of our interest which can be used to extract the information about the eigenstate fractal dimensions (8). Like in the paper for the RP model [15], we would like to focus on the statistics of the local resolvent $G(z) = (z - H)^{-1}$ with $z = \varepsilon - i\eta$. Following [15], in order to probe the non-ergodic properties of the system we consider $\eta = N^\alpha \delta \sim 1/N^{1-\alpha}$, $\alpha > 0$, to be much larger than the mean level spacing $\delta$ determined by the diagonal disorder.[6] In this case we are able to detect an arbitrary small non-zero fractal dimension by choosing a finite positive constant $\alpha < D(\beta)$. This works when the global resolvent $\mathrm{Tr}\,[G(z)]/N$ (the imaginary part of which is just the global density of states $\rho \sim (N\delta)^{-1}$) is a featureless function converging to an order-one number in the thermodynamic limit $N \to \infty$.[7] For the opposite case, please see in Appendix B.

The local density of states (LDOS) given by the imaginary part of the resolvent's diagonal element $G_{ii}(z)$ has the form

$$\frac{1}{\pi}\,\mathrm{Im}\,G_{ii}(z) = \sum_{n=1}^{N} \frac{\eta/\pi}{(\varepsilon - E_n)^2 + \eta^2}|\psi_n(i)|^2\,, \tag{12}$$

where $\psi_n(i) = \langle i|\psi_n\rangle$ is the eigenstate's amplitude on site $i$; the sum represents a weighted average of $|\psi_n(i)|^2$ over the energy window $(\varepsilon - \eta, \varepsilon + \eta)$ and can be linked to the spatial structure of the eigenstates in this window. Now, consider an ensemble of Hamiltonians $H$ with different hopping matrices $V$ but fixed $H_0$. If we assume the close in energy eigenstates from *different realizations* to live on almost *the same fractal*, see the bottom panel of Fig. 3, we can extract the information about the broadening $\Gamma$ of the Lorenzian, determining the fractal dimension (8), from the resolvent averaged over the hopping disorder (please see Eqs. (30, 31) for more details)

$$\overline{G}(z) = \left\langle \frac{1}{z - H_0 - V} \right\rangle_V \simeq \frac{1}{z - H_0 - \Sigma}, \text{ with } \mathrm{Im}\,\Sigma \equiv \Gamma \sim N^{D(\beta)}\delta\,. \tag{13}$$

This last assumption looks reasonable when the energy landscape of $H_0$ plays a dominant role in forming of the eigenstates profile, e.g., at least in case with all hopping elements of roughly the same amplitude. Hence, by choosing $\overline{G}(z)$ as our observable, we are restricting ourselves to such models.

## 4 Previously known methods

In this section we consider a couple of previously known methods and explain why they are not applicable for the models of our interest.

The cavity method is based on the block-matrix inversion formula

$$G_{ii}(z) = \left(z - \varepsilon_i - \sum_{j,k \neq i} V_{ij} G_{jk}^{(i)}(z) V_{ki}\right)^{-1}, \tag{14}$$

for the diagonal part of the local resolvent written in terms of the resolvent $G_{jk}^{(i)}(z)$ of the Hamiltonian with the $i$th row and column being removed (see Fig. 4(a)). Both for the tree-like

---

[6]For $\varepsilon$ lying outside of the spectral bulk the expression should be modified as $\eta(\varepsilon) = N^\alpha \delta(\varepsilon)$, where $\delta(\varepsilon)$ is a mean level spacing at a certain energy $\varepsilon$.

[7]The more restrictive condition for $\alpha$ to be infinitesimally small but positive makes it possible to detect $D(\beta) > 0$ for any model regardless of the global resolvent behaviour.

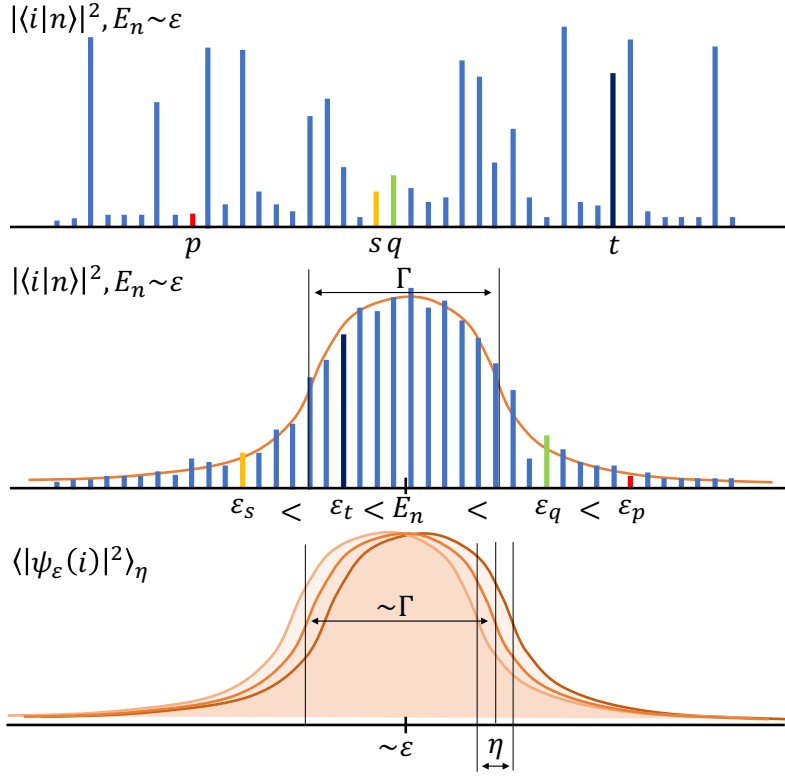

Figure 3: **Averaging over the energy window**. (Upper panel) Sketch of the wave function spatial distribution, $|\psi_\varepsilon(i)|^2$ vs $i$, for a certain disorder realization. (Middle panel) The same $|\psi_\varepsilon(i)|^2$ plotted versus the diagonal energies $\varepsilon_{i_k}$, reordered in the increasing order of $\varepsilon_{i_1} < \varepsilon_{i_2} < \ldots < \varepsilon_{i_N}$. The corresponding profile shows Lorenz-like behavior around the energy $E_n$ with the width given by the fractal dimension as $\Gamma \sim N^D \delta$. (Lower panel) The local resolvent, Eq. (12) and its average over off-diagonal elements, Eq. (13), show more or less the same Lorenz-like profile with the width $\Gamma$ provided $\eta \ll \Gamma$ and the close-in-energy eigenstates live on close fractals.

graphs [70, 71] and RP-like models [15, 24] in the thermodynamic limit $N \to \infty$ it is known that the contribution of off-diagonal elements $G_{j \neq k}^{(i)}(z)$ is negligible giving the self-consistent equation for the diagonal $G_{ii}(z)$ only:

$$G_{ii}(z) = \left( z - \varepsilon_i - \sum_{j \neq i} |V_{ij}|^2 G_{jj}^{(i)}(z) \right)^{-1}. \tag{15}$$

This method works well for the dense matrices with the uncorrelated entries $V_{ij}$ [15, 19]. Indeed, in this case $G_{jk}^{(i)}(z)$ is independent from $V_{ij}$ and $V_{ki}$ for any $j$, $k$ and the sum in r.h.s. can be tackled using the central limit theorem. For $N \to \infty$ the local resolvent averaged over $V_{ij}$ takes the form

$$\overline{G}_{ii}(z) = \left( z - \varepsilon_i - \text{Tr}\left[ \overline{G}^{(i)}(z) \right] \left\langle |V_{ij}|^2 \right\rangle \right)^{-1}. \tag{16}$$

However, in the case of the correlated $V_{ij}$ (5) it is hard to proceed beyond Eq. (15).

Another method used for the description of the RP model is the Dyson Brownian motion approach [15]. It is based on the stochastic process of adding random Gaussian matrices $dV(t)$ to the special diagonal $H_0$ (see Fig. 4(b))

$$H(t + dt) = H(t) + dV(t), \quad H(0) = H_0, \quad H(1) = H. \tag{17}$$

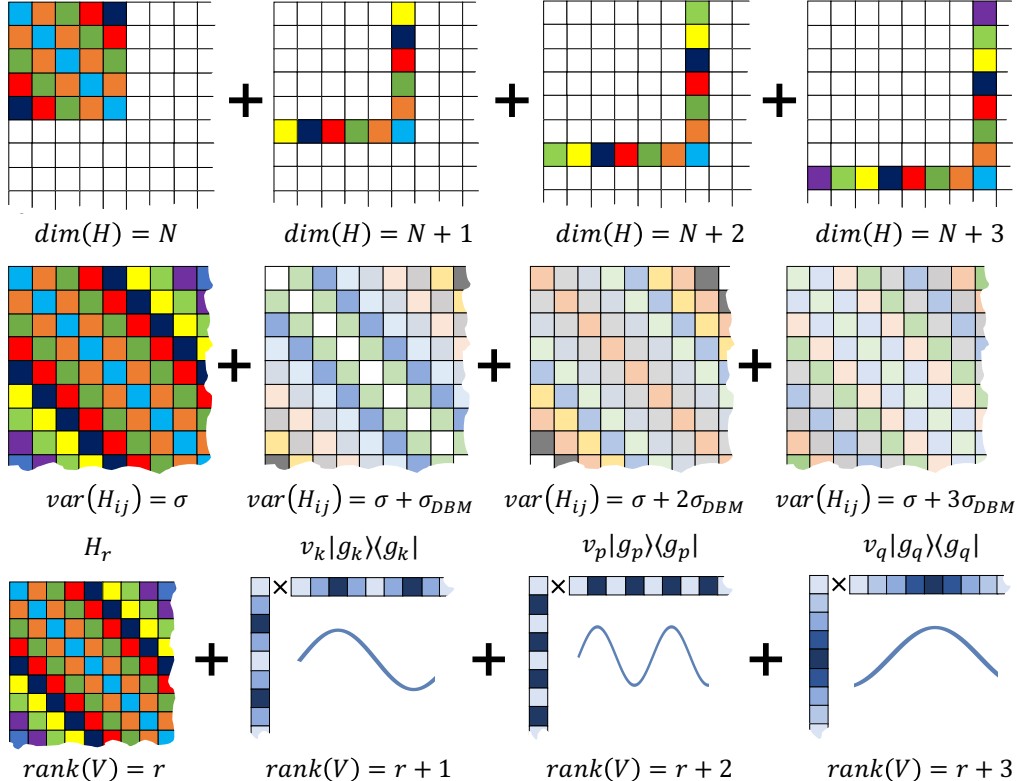

Figure 4: A comparison of iteration schemes for TIRP model behind (a) the cavity method (when at each iteration step one column and one row are added, i.e. increasing the matrix size by one), (b) the Dyson Brownian motion method (when at each step the random matrix of the same form but with a much smaller entries' variance $\sigma_{DBM}$ is added), and (c) the rank-one increment method (when at each step a rank-one matrix of the form $v|g\rangle\langle g|$ is added).

This method allows to calculate the instant eigenvectors and eigenvalues of the problem from the stochastic differential equations and even write a closed equation for the local resolvent $G_{ii}(z)$ averaged over the *uncorrelated Gaussian* off-diagonal elements $V_{ij}$ [15]

$$\partial_t \overline{G}_{ii}(z,t) = -N \left\langle |V_{ij}|^2 \right\rangle \overline{G}_{ii}(z,0) \partial_z \overline{G}_{ii}(z,t) \,. \tag{18}$$

Like the cavity method, the Dyson Brownian motion approach is limited by the statistical independence of the additions $dV_{ij}(t)$ to the matrix elements both at different time instants and different coordinates $(i,j)$.

## 5 The rank-one increment method

### 5.1 The main idea

In order to overcome the problem with the correlated character of the hopping matrix entries $V_{ij}$ we consider the matrix $V$ as a sum of uncorrelated rank-one matrices. For example, we can use a natural eigenbasis decomposition (1b). Then we get a series of Hamiltonians

$$H_r = H_{r-1} + v_r |g_r\rangle\langle g_r| = H_0 + \sum_{k=1}^{r} v_k |g_k\rangle\langle g_k| = H_0 + V_r \,, \tag{19}$$

with $H_0 = H_0$ ($V_0 = 0$) and $H_R = H$. Exploiting the so-called Sherman-Morrison formula, one obtains a corresponding series of resolvents

$$G_r(z) = G_{r-1}(z) + G_{r-1}(z)S(v_r, g_r, G_{r-1}(z))G_{r-1}(z), \tag{20a}$$

$$S(v, g, G) = \frac{v|g\rangle\langle g|}{1 - v\langle g|G|g\rangle}. \tag{20b}$$

We use this exact recursion to obtain a differential equation for the resolvent $\overline{G}(z; r)$ averaged over the off-diagonal disorder. Note that in the case of uncorrelated hopping entries this equation resembles the aforementioned Dyson Brownian motion approach result.

The graphical representation of the method is given by Fig. 4(c). Unlike two previous methods, this one does not suffer from the correlated nature of the added matrices and uses the natural decomposition of energy stratified models. Apart from an excellent quantitative description of the model's characteristics, the resulting differential equations give deep insights into how exactly the fractal phase emerges with lifting more and more hopping eigenvalues from zero.

## 5.2 Self-averaging

A key point which allows us to proceed with the above method is a statement about self-averaging of a quantity $\langle g|G|g\rangle$ which enters a denominator of (20b).[8] Indeed, if this statement fails, the r.h.s. of (20a) becomes highly nonlinear and extremely difficult to tackle analytically while in case of aforementioned self-averaging the matrix $S$ depends only linearly on $|g\rangle\langle g|$ and doesn't depend on $G$ at all.

To get an idea when and why this self-averaging condition holds, let us write the considered matrix element explicitly in the eigenbasis $\{|n\rangle\}$ of $H_{r-1}$:

$$\langle g_r|G_{r-1}(z)|g_r\rangle = \sum_{n=1}^{N} \frac{|\langle n|g_r\rangle|^2}{\varepsilon - i\eta - E_n}. \tag{21}$$

Due to the imaginary part $\eta$ of $z$, the main contribution to this sum goes from $\sim \eta/\delta \sim N^\alpha$ states in a vicinity of the energy $\varepsilon$ (assuming that we are in the spectral bulk). Then if the corresponding components of $|g\rangle$ are "ergodic", i.e., all $|\langle n|g\rangle|^2$ are of the same order $1/N$, then it is clear that the whole sum must only slightly fluctuate around some definite value of the order of unity. The above "local ergodicity" condition is likely to hold until the corresponding fractal dimension of $|n\rangle$ (with $|E_n - \varepsilon| \lesssim \eta$) in the basis $|i\rangle$ is less than one while for $|g\rangle$ the fractal dimension is assumed to be equal to one. As a result, we come to a conclusion that this self-averaging holds at least as long as we work *in the spectral bulk* and in one of the *non-ergodic phases*. In other words, we claim that in the aforementioned cases (21) can be replaced by

$$\langle g|G(z)|g\rangle \to \frac{1}{N}\text{Tr}[G(z)] \simeq i\pi\rho. \tag{22}$$

In the last equality we drop the real part of this quantity out due to its unimportance to the localization effects, and $\rho = 1/(N\delta)$ is a global density of states. For more details on the self-averaging derivation (complimented by the numerical evidences) please look into the Appendix A.

Substituting (22) into (20), we obtain the self-averaged version of the Sherman-Morrison formula

$$G_r(z) = G_{r-1}(z) + G_{r-1}(z)\frac{v_r|g_r\rangle\langle g_r|}{1 - i\pi\rho v_r}G_{r-1}(z). \tag{23}$$

---

[8]In some sense it is similar to the derivation of Eq. (16) from (15) in the cavity method.

This formula is somehow similar both to the cavity method on the tree (15) (but non-linear in $G_{r-1}(z)$)) and to the Dyson Brownian Motion approach (18).

In next sections we consider separately two distinguished cases of

**(i)** $\beta < 1$ with the measure zero of non-zero energy levels in the hopping term, and

**(ii)** $\beta = 1$ with the finite fraction of such levels.

This separation is dictated by the following fact. The averaging over the hopping matrix $V$, Eq. (13), contains not only the averaging over its i.i.d. eigenvalues $v_r$ (4), but also over eigenvectors $|g_r\rangle$. According to Eq. (23) this averaging can be done step by step over $v_r$ and $g_r$ with increasing index $1 \le r \le N^{\beta}$. However, at each $r$th step the averaging over $|g_r\rangle$ should imply the orthogonality of this vector to all the previous ones $|g_{k<r}\rangle$.

For $\beta < 1$ the measure of such vectors is zero among the overall basis $|g_m\rangle$ and therefore their effect on the averaging is negligible in the thermodynamic limit. This leads us to the simple results from Sec. 5.3 analogous, but not equivalent to the ones in the uncorrelated RP model [15].

Unlike this for $\beta = 1$ the orthogonality of the vectors $|g_r\rangle$ play a significant role and might make the results different in the case $\beta = 1$ with respect to the one given by a limit $\beta \to 1$.[9] Therefore in Sec. 5.5 we develop the method combining the Sherman-Morrison formula (23) with the Dyson Brownian motion.

## 5.3 Cavity-like method for an intermediate regime $\beta < 1$.

Using the arguments given in the previous section for the case of $\beta < 1$ we average Eq. (23) over independent $v_r$ and $g_r$ with fixed $r$ and obtain the recursive equation similar to the one of the cavity method on the tree structure (15)

$$\langle G_r(z) \rangle_{\{v_r, g_r\}} = G_{r-1}(z) + \sigma G_{r-1}(z)^2, \quad \sigma = \frac{1}{N} \int \frac{v p(v) \mathrm{d}v}{1 - \mathrm{i}\pi\rho v} . \tag{24}$$

Here the integral in $\sigma$ is given by the averaging over the probability distribution $p(v)$ of the hopping eigenvalues $v$, while the averaged projector $|g_r\rangle\langle g_r|$ gives the unity matrix multiplied by $1/N$.[10]

In order to proceed with the averaging over all other $g_k$ and $v_k$ with $k < r$, one can replace the squared resolvent by the derivative $G(z)^2 \equiv -\partial_z G(z)$. After this trick the equation becomes linear in $G(z)$ and can be straightforwardly averaged over $g_k$ and $v_k$ with $k < r$

$$\overline{G_r}(z) = \overline{G_{r-1}}(z) - \sigma \partial_z \overline{G_{r-1}}(z). \tag{25}$$

Finally, assuming that $\sigma \overline{G_{r-1}}(z)$ is sufficiently small, we arrive at the desired differential equation for the mean resolvent:

$$\partial_r \overline{G}(z; r) + \sigma \partial_z \overline{G}(z; r) = 0. \tag{26}$$

---

[9]For example, consider a model with $V = cI = \sum_{k=1}^{N} c|g_k\rangle\langle g_k|$, i.e. with $V$ being a non-trivial hopping term at any $\beta < 1$, but collapsing to just a finite energy shift at $\beta = 1$. It is clear that in this case $D(\beta = 1)$ is zero while the $\beta \to 1$ limit of $D(\beta)$ is one. This example illustrates a highly correlated case when the aforementioned orthogonality really matters.

[10] One may argue that for $r > 1$ the projector $|g_r\rangle\langle g_r|$ cannot be averaged to identity matrix due to the orthogonality conditions $\langle g_p|g_q\rangle = \delta_{pq}$. However, for $\beta < 1$, a number of the orthogonality conditions is of measure zero, and we will neglect their effect here and after. So, as the title of the current subsection implies, this result holds only for $\beta < 1$, but we will see below in Sec. 5.5 how it can be generalized to a full-rank case.

From this differential equation it is apparent that $\overline{G}(z;r) = F(z - \sigma r)$ is a function of the propagating wave argument $z - \sigma r$. With the initial condition $\overline{G}(z;0) = (z - H_0)^{-1}$, we get the resolvent averaged over the hopping in the form

$$\overline{G}(z,R) = (z - H_0 - \Sigma(R))^{-1}, \quad \Sigma(R) = \sigma R, \quad R = N^\beta, \quad \beta < 1. \tag{27}$$

From the result one can see that the assumption of the smallness of $\sigma \overline{G_{r-1}}(z)$ is valid until $\sigma \ll 1$.

The recursive equation (25) has much in common with the cavity method (15) as well as Eq. (26) is similar to the one of the Dyson Brownian motion (18). Even the result (27) for $\beta \to 1$ is formally the same as the one of the cavity method (16). However unlike these methods both equations (25), (26) are derived for the correlated matrix elements $V_{ij}$ for which both cavity and DBM methods fail. In any case we nickname this method "cavity-like" due to the similarity between recursive equations.

## 5.4 A fractal dimension $D$

As it is apparent from the results of the previous section, it is only the parameter $\sigma$ which matters for the determination of the self-energy $\Sigma$ for the models (1) with $\beta < 1$ and independently distributed eigenvalues $v_k$, and

$$\sigma = \frac{1}{N}\left( \int \frac{vp(v)\mathrm{d}v}{1 + \pi^2\rho^2 v^2} + \mathrm{i}\pi\rho \int \frac{v^2 p(v)\mathrm{d}v}{1 + \pi^2\rho^2 v^2} \right). \tag{28}$$

The real part of $\sigma$ results in the energy shift. For the symmetric distribution $p(v)$ it vanishes, while in general for the mid-spectrum energies $\varepsilon$, it can be neglected provided $\mathrm{Re}\,\Sigma(\beta) = \mathrm{Re}\,\sigma N^\beta \ll 1$ is less than the bandwidth. Assuming this is the case, let's focus on the imaginary part which then determines the localization properties of the wave functions. The imaginary part, in turn, can be estimated up to the unimportant prefactor as

$$\mathrm{Im}\,\sigma = \frac{\pi\rho}{N} \int \frac{v^2 p(v)\mathrm{d}v}{1 + \pi^2\rho^2 v^2} \sim \min\left\{ \frac{\rho}{N}\left\langle v^2 \right\rangle, \delta \right\}, \tag{29}$$

where the first term in r.h.s. corresponds to the small $\left\langle v^2 \right\rangle \ll 1/\rho^2$, while the second gives the cutoff for the large values $\left\langle v^2 \right\rangle \gg 1/\rho^2$; $\delta = 1/(\rho N)$ is the mean level spacing in the spectral bulk.

Now, recalling the argument about the average site population following (12) and taking the imaginary part of (27), we get

$$\left\langle |\psi_\varepsilon(i)|^2 \right\rangle_V = \frac{1}{N} \frac{\mathrm{Im}\,\Sigma(\beta)}{(\varepsilon - \varepsilon_i - \mathrm{Re}\,\Sigma(\beta))^2 + (\mathrm{Im}\,\Sigma(\beta))^2}. \tag{30}$$

Together with (29) and the standard definition of the fractal dimension $D$ via the support set of the wave function

$$N^D \sim \mathrm{Im}\,\Sigma/\delta \sim N^{1+\beta}\,\mathrm{Im}\,\sigma, \tag{31}$$

the Eq. (30) finally gives

$$D(\beta) = \beta + \log_N \min\left\{ \rho\left\langle v^2 \right\rangle, 1/\rho \right\}. \tag{32}$$

Substituting particular values (4), $\left\langle v^2 \right\rangle = N^{2-\gamma-\beta}$, corresponding to our model (1) at $\gamma < 2$, we get

$$D(\beta) = \begin{cases} \beta, & 0 < \beta < 2 - \gamma, 1 \\ 2 - \gamma, & 2 - \gamma < \beta < 1 \end{cases}, \tag{33}$$

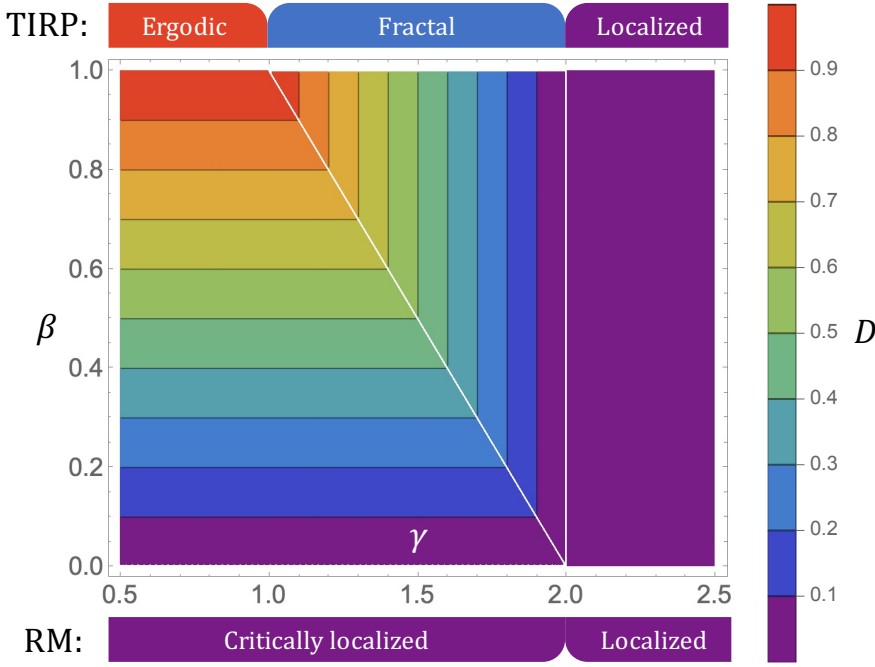

Figure 5: A phase diagram (33) of a model (1) with hopping rank $\beta$ and scaling parameter $\gamma$. Two limiting cases, Richardson's model ($\beta = 0$) and TIRP ($\beta = 1$) are highlighted.

restoring the dependence (11) already shown in Fig. 2. The overall phase diagram in the plane $(\beta, \gamma)$ is given by Fig. 5. It shows how the Richardson's model (noted as RM) looses its localization properties at $\gamma < 2$ as soon as one adds an extensive number $N^{\beta > 0}$ of energy stratified hopping terms.

Indeed, for any fixed $0 < \beta < 1$ the localized phase at $\gamma < 2$ is replaced by the fractal one with the maximal fractal dimension given by $D = \beta$, see Fig. 5. This maximal fractal dimension can be physically understood as follows. The $N - N^{\beta}$ low-energy eigenstates of the translation-invariant model (without the diagonal disorder $H_0$) are degenerate and therefore should transform to the delta-function localized ones in the coordinate basis as soon as one adds the diagonal disorder $H_0$ to the problem. However this delta-function localization is limited by the presence of $N^{\beta}$ energy-stratified states. It is straightforward to show that these $N^{\beta}$ states are localized in the momentum $\{|g_k\rangle\}$ basis due to their extensive energies, while all the rest states should be orthogonal to them. Considering also that the hybridized low-energy states should be orthonormal to each other, we get that such a state typically occupies at least $N^{\beta}$ sites. This finally puts the minimal support set $N^{\beta}$ of the mid-spectrum states in our models and provides the necessary understanding of the result for the fractal dimension $D = \beta$ instead of delta-peak localization with $D = 0$, see Fig. 1(e). The above result works for $\gamma < 2 - \beta$ as soon as the high-energy states are energetically stratified from the bulk of the spectrum, Fig. 1(a-b).

On the other hand, in the interval of $2 - \beta < \gamma < 2$ the fractal dimension is given by $D = 2 - \gamma$ as in the TIRP model, see Fig. 5. This corresponds to the bandwidth $\langle v^2 \rangle \sim N^{2-\gamma-\beta}$ of the high-energy states to be of the order of the diagonal disorder or smaller. As a result, in such a case these $N^{\beta}$ levels are not anymore energy-stratified and the whole picture goes back to the TIRP case which practically shows the same result as the Fermi's golden rule consideration.

The above result in the limit $\beta \to 1$ is also consistent with the previously known results for

$\beta = 1$ despite the fact that this derivation is formally applicable only for $\beta < 1$. In the next subsection we make sure that this is not a coincidence.

## 5.5 Dyson-Brownian-motion-like stochastic process for TIRP: $\beta = 1$

Now, consider the full-ranked hopping matrix $V$ of size $N \times N$. As it was mentioned in the end of Sec. 5.2, the averaging over $|g_r\rangle$ in the form presented in Eq. (24) doesn't work for $r \sim N$ because during the averaging we have to exclude the subspace spanned by $|g_k\rangle$ with $k < r$, and such effects cannot be neglected anymore. It provides some correlations between the averaging on the current step and all the previous ones and makes the analysis much more involved.

In order to proceed, in the full-ranked version of our model (1) we use the analogy between the recursive representation (19) and the one of the Dyson Brownian motion (17). Indeed, like in the Dyson Brownian motion, one can consider a stochastic random matrix process over auxiliary time, $0 \le t \le 1$, defined as a $\Delta t \to 0$ limit of the recursion (20a)

$$H(t + \Delta t) = H(t) + \Delta v |g\rangle\langle g|, \quad H(0) = H_0, \tag{34}$$

where the random eigenvalue increment $\Delta v \sim \sqrt{N \Delta t}$ and the random wavevector $g = \overline{1, N}$ are chosen *independently* on each step. It is important to emphasize that here, unlike the cavity-like method case, Sec. 5.3, the parameter $\Delta v$ is not given by eigenvalue of the matrix $V$, but it should be found in such a way to sample the desired matrix ensemble at $t = 1$. This, in particular, means that each individual eigenvector $|g_k\rangle$ appears extensive amount of times $\sim (N \Delta t)^{-1} \gg 1$ (but not once) during the random process. In order to emphasize this we everywhere use $N \Delta t \ll 1$ as a small parameter. Hence, since this approach doesn't force the eigenvector $|g\rangle$ added on some $k$'th step to be orthogonal to all previously added ones, the aforementioned orthogonality problem is not a problem anymore, and we can use the analogue of the averaged Sherman-Morrison formula (23) with $v_k$ replaced by $\Delta v_k$ for each individual step of the stochastic matrix process.

In the following, we focus on the $\beta = 1$ TIRP model which is characterized by the Gaussian distribution of the eigenvalues (4). This immediately simplifies our stochastic matrix process to a process with the Gaussian eigenvalue increments $\Delta v = v\sqrt{N \Delta t}$, $v \sim \mathcal{N}(0, N^{-\gamma})$. Now, using the self-averaged version (23) of the Sherman-Morrison formula and expanding it analogously to the Dyson Brownian motion in powers of $N \Delta t$ up to the first power, one obtains

$$G_{t+\Delta t}(z) = G_t(z) + \left( v\sqrt{N \Delta t} + i\pi\rho(v\sqrt{N \Delta t})^2 + ... \right) G_t(z) |g\rangle\langle g| G_t(z). \tag{35}$$

With all the above simplifications given by the random process, one averages the latter equation in the same way as (24) and derives the analogous equation

$$\partial_t \overline{G}(z; t) + i\pi\rho \left\langle v^2 \right\rangle \partial_z \overline{G}(z; t) = 0. \tag{36}$$

From this equation one obtains

$$\Sigma(t) = i\pi\rho \left\langle v^2 \right\rangle t. \tag{37}$$

Recalling that the desired full-ranked model arises from our stochastic process at $t = 1$ we finally get $\Sigma(\beta = 1) = i\pi\rho N^{1-\gamma}$ which gives similarly to the previous section $D = 2 - \gamma$ as expected [63].

One may notice the similar paradigm of this section as in the Dyson Brownian motion, therefore we use this name of "DBM-like" for this method. However, like in Sec. 5.3, the equivalence between Eq. (36) and (18) and formally the same results should not confuse a reader as they are derived for the TIRP model with translation-invariant correlations of the matrix elements $V_{ij} = V_{i-j}$ for which the DBM method does not work.

# 6 Conclusions

To sum up, in this paper we consider the set of disordered random matrix models with translation-invariant correlations in the hopping term. This set of models interpolates between the completely correlated case of the Richardson's model and the translation-invariant version of the Rosenzweig-Porter (TIRP) model. The former model is Bethe ansatz integrable and known to have all (except one) eigenstates to be localized even beyond the convergence of the locator expansion, while TIRP model shows the same fractal wave-function statistics as the usual uncorrelated RP one. The interpolation between these models is given by a number $N^\beta$, $0 \le \beta \le 1$, of finite-energy levels in the spectrum of the translation-invariant hopping in the combination with the energy-degenerate rest eigenstates of the disorder-free translation-invariant model. Such kind of the energy stratification is known to play an important role in cooperative shielding and correlation-induced localization.

However quite surprisingly in contradiction to the cooperative shielding arguments, we have found that as soon as the number of energy-stratified levels starts to grow extensively with the system size $N$, the mid-spectrum states become delocalized beyond the locator expansion, but fractal.

In order to describe the above set of models and calculate the fractal dimension of the bulk spectral states, we develop the generalizations of the cavity and the Dyson Brownian motion methods, based on the natural spectral decomposition of the correlated hopping matrix complemented by the Sherman-Morrison formula for the local resolvent.

Our methods uncover the whole phase diagram of the above-mentioned set of models and shows that for all $0 < \beta < 1$ the localization beyond the locator expansion, which is present in the Richardson's model, is replaced by the emergent fractal phase in the same parameter range.

The numerical simulations check the validity of the above methods and confirm the analytical results for the fractal dimensions. The first of these methods has a wider application than just a Gaussian distribution of the eigenvalues that will be considered in more detail in further publications.

# 7 Outlook and discussions

As a next step in the development of the approach suggested in this paper, one can consider the cases of either or both eigenvalues $v_k$ and eigenvectors $g_k$ of the hopping matrix to be correlated between each other. This direction has immediate applications in the description of short-range graph models and their mapping to the random-matrix ensembles (see, e.g. [49]).

Another promising direction is to consider the fat-tailed distribution of eigenvalues $v_k$ in the full-rank case, similarly to uncorrelated models [22–24, 49] and uncover the possible nature of the emergent multifractality for such models.

This is the subject of further investigations, but here we would like to notice that the DBM-like method developed in Sec. 5.5 cannot be immediately applied for the fat-tailed distributed $v_k$. Indeed, as soon as the PDF of $\Delta v$ is not narrow and the fluctuations are comparable with the bandwidth, $\langle \Delta v^2 \rangle - \langle \Delta v \rangle^2 \gtrsim \rho^{-2}$ the entire self-averaging argument breaks down. An origin of this fact lies in a behaviour of $\langle g_k|G|g_k \rangle$ on later steps of the random process when the state $|g_k\rangle$ was already used earlier. This quantity can be computed exactly using (20a) and looks like

$$\langle g_k|G_{r+1}|g_k \rangle = \frac{\langle g_k|G_r|g_k \rangle}{1 - v_k(t)\langle g_k|G_r|g_k \rangle} \sim \frac{i\pi\rho}{1 - i\pi\rho v_k(t)}, \tag{38}$$

where $v_k(t)$ is the corresponding hopping matrix eigenvalue on step $t$. Since $v_k(t)$ is not zero for the later stages of the process, it can break the convenient self-averaging property of $\langle g_k | G_{r+1} | g_k \rangle$ and, if the fluctuations of $\Delta v_k$ (and, hence, $v_k(t + \Delta t)$) are large, we cannot neglect this effect. As a result, $S(v, g, G)$ no longer can be naively averaged to a multiple of unity independent of $G$ which leads to significant complications and lies beyond the scope of the present paper.

## Acknowledgements

**Funding information** This work was supported by Russian Science Foundation (Grant No. 21-12-00409).

## A Self-averaging

Consider the matrix element $\langle g_r | G_{r-1}(z) | g_r \rangle$ in the denominator of the fraction from (20b). We claim that this quantity self-averages almost always for a range of parameters we are interested in. This claim is based on the assumption that the added on the $r$'th step hopping eigenstate $|g_r\rangle$ is *generic* with respect to the eigenstates $|n_{r-1}\rangle$ of the Hamiltonian $H_{r-1}$ from the previous step, i.e., that its fractal dimension in the basis of the Hamiltonian's eigenstates is one. And, if $|g_r\rangle$ is ergodic in the basis of $H_0$ than this is indeed the case at least until $|n_{r-1}\rangle$ are *not ergodic* in that basis, i.e. until $D(r) < 1$. The only obvious thing which can break this logic is the orthogonality condition $\langle g_r | g_p \rangle = \delta_{rp}$ between $|g_r\rangle$ and $|g_p\rangle$ when $|g_p\rangle$ was added on the previous steps of the recursion (20a) thereby making $|n\rangle$ and $|g_r\rangle$ correlated. As we know from the analysis of the Richardson model, for a very large eigenvalue $v_p$, the corresponding detached eigenstate will be almost collinear to $|g_p\rangle$ and, hence, indeed orthogonal to all other $|g_r\rangle$ with $r \neq p$. However, as long as this happens only for *detached* levels, there is no reason to think about such effects if we talk about bulk states.

So now, when we agreed to work *in the spectral bulk* and *in a non-ergodic phase*, we can approximate $\langle n | g_r \rangle$ as taken from the normal distribution and write for $\langle g_r | G_{r-1}(z) | g_r \rangle$ and some fixed $G_{r-1}(z)$ that

$$\langle g | G(z) | g \rangle \overset{dist}{\approx} \tau(z) = \frac{1}{N} \sum_{n=1}^{N} \frac{x_n^2}{z - E_n} , \tag{39}$$

where $x_n$ are independently distributed standard normal random variables which, together with the factor $1/N$, approximate the uniform over the unitary group distribution of $|g\rangle$ in the large-$N$ limit [72].

The distribution of $\tau(z)$ can be estimated by computing separately the characteristic functions of its real and imaginary parts. For example,

$$Q_{\text{Im}}(\xi) = \left\langle e^{i\xi \, \text{Im} \, \tau(z)} \right\rangle = \exp \left\{ -\frac{1}{2} \sum_{n}^{N} \ln \left( 1 - \frac{2i\xi}{N} \frac{\eta}{(\varepsilon - E_n)^2 + \eta^2} \right) \right\} . \tag{40}$$

Then, taking $\delta(\varepsilon) \ll \eta \ll \rho(\varepsilon)/\rho'(\varepsilon)$, where $\delta(\varepsilon) = 1/N\rho(\varepsilon)$ is the energy-dependent mean level spacing and $\rho(\varepsilon)$ is the density of states, we can approximate the sum in the exponent by the integral

$$I(\xi) = \int \ln \left( 1 - \frac{2i\pi\xi}{N} \frac{1}{\pi} \frac{\eta}{(\varepsilon - E)^2 + \eta^2} \right) N\rho(E) \mathrm{d}E . \tag{41}$$

This integral can be approximately calculated for a fixed $N$ in two limiting cases: in the case $\xi \ll N\eta$ when the logarithm can be expanded in the Taylor series and in the case $\xi \gg NE_{BW}^2/\eta$ where $E_{BW}$ is the bandwidth of the Hamiltonian. The desired asymptotics then take the form

$$I(\xi) \sim \begin{cases} -2i\pi\xi\rho(\varepsilon), & \xi \ll N\eta \\ N\ln(-2i\eta\xi/N) - NC(\varepsilon,\eta), & \xi \gg NE_{BW}^2/\eta \end{cases}, \tag{42}$$

where

$$C(\varepsilon,\eta) = \int \ln\left(\eta^2 + (\varepsilon - E)^2\right)\rho(E)\mathrm{d}E, \tag{43}$$

is a constant independent of $\xi$. Now, recalling from Sec. 5.1 that the proper choice of $\eta$ is $\eta(\varepsilon) = \delta(\varepsilon)N^\alpha$ with $\alpha > 0$, we see that the region where the characteristic function $Q_{\mathrm{Im}}(\xi)$ oscillates with a constant amplitude grows with the system size $N$, and beyond this region it decays polynomially with the power of $N/2$. All this points to the fact that the distribution of $\mathrm{Im}\,\tau(z)$ becomes narrower with the increasing system size and concentrates around the mean point $\langle \mathrm{Im}\,\tau(z) \rangle = \pi\rho(\varepsilon)$. The distribution function of $\mathrm{Re}\,\tau(z)$ behaves in a similar way concentrating around $\pi\partial_\varepsilon C(\varepsilon,\eta)$, and this finally justifies our self-averaging approximation giving

$$\langle g|G(z)|g \rangle \to \frac{1}{N}\mathrm{Tr}\left[G(z)\right], \tag{44}$$

until we work in the spectral bulk and beyond the ergodic phase. The second and the fourth rows of Figure 6 demonstrate the self-averaging of the corresponding distributions.

It is also worth mentioning that, in the limit $\eta \lesssim \delta(\varepsilon)$, one cannot transform the sum, Eq. (40), into the integral, Eq. (41), since particular values of $E_n$ now matter. Instead of a single frequency $2\pi\rho(\varepsilon)$, $Q_{\mathrm{Im}}(\xi)$ has now a set of equally important realization-dependent frequencies and, hence, no self-averaging, see the first and the third rows of Figure 6.

Finally, since $\langle \mathrm{Re}\,\tau(z) \rangle = \pi\partial_\varepsilon C(\varepsilon,\eta)$ for the bulk energies is of the order of the bandwidth or even parametrically smaller than that, we drop this real part and write just $\langle g_r|G_{r-1}(\varepsilon - i\eta(\varepsilon))|g_r \rangle \sim i\pi\rho(\varepsilon)$ which will simplify the calculations without qualitatively affecting the results or losing the generality.

## B  A self-consistent approach

In the case of not smooth density of states, one cannot drop the energy dependence from the self-averaging matrix element $\langle g_r|G_{r-1}(z)|g_r \rangle$. In such a case one should consider the self-consistent formulation of the problem. In case of $\beta < 1$ it can be written as

$$\begin{cases} \partial_r \overline{G}(z,r) + \sigma(z,r)\partial_z \overline{G}(z,r) = 0 \\ \sigma(z,r) = \frac{1}{N}\int \frac{vp(v)\mathrm{d}v}{1 - v\mathrm{Tr}\left[\overline{G}(z,r)\right]/N} \end{cases}, \tag{45}$$

where $\sigma$ now depends both on the energy $z$ and the rank $r$. In case of Dyson Brownian motion-like approach for $\beta = 1$ it is ideologically similar and differs only by a definition of $\sigma$ and "time" $r$. The self-consistency condition can be extracted from this system and treated independently: after taking a trace of the first equation and defining $\overline{\tau}(z,r) = \mathrm{Tr}\left[\overline{G}(z,r)\right]/N$ we get a closed non-linear system

$$\begin{cases} \partial_r \overline{\tau}(z,r) + \sigma(\overline{\tau})\partial_z \overline{\tau}(z,r) = 0 \\ \sigma(\overline{\tau}) = \frac{1}{N}\int \frac{vp(v)\mathrm{d}v}{1 - v\overline{\tau}} \end{cases}, \tag{46}$$



**Figure 6:** Probability distribution functions of the random quantity $\mathrm{Re}\langle g|G(z)|g\rangle$ (left panels) and $\mathrm{Im}\langle g|G(z)|g\rangle$ (right panels) for the model (1) in the middle of the spectrum for different system sizes (shown together with the number of disorder realizations in the legend). The scaling of the broadening parameter $\eta$ and the number of ranks $R$ taken are (top panels) $\eta \sim N^{-1.1}$, $R = N^{1/2}$, (middle-top panels) $\eta \sim N^{-0.9}$, $R = N^{1/2}$, (middle-bottom panels) $\eta \sim N^{-1.1}$, $R = N - N^{1/2}$, and (bottom panels) $\eta \sim N^{-0.9}$, $R = N - N^{1/2}$. The insets show the standard deviation of the distributions versus the system size.

and the equation for $\overline{G}(z,r)$ then gives $\overline{G}(z,r) = \overline{G}_0(z - \sigma(\overline{\tau})r)$ with the self-consistently determined $\sigma(\overline{\tau}(z,r))$. In turn, the system (46) can be easily solved using the method of characteristics which for the initial conditions $\overline{\tau}(z,0) = \overline{\tau}_0(z)$ gives the solution in the form

$$\overline{\tau}(z,r) = \overline{\tau}_0(z - \sigma(\overline{\tau})r) = \overline{\tau}_0\left(z - r\sigma\left(\overline{\tau}_0\left(z - r\sigma\left(\overline{\tau}_0(\ldots)\right)\right)\right)\right). \tag{47}$$

### B.1 Order-independence property

In general, there are $R!$ different ways to obtain the same $G(z)$ using the exact recursion (20a); the ways differ by the order we choose to lift the hopping eigenvalues $v_k$ from zero. It is reasonable now to check if our self-consistent equation (45) for the mean resolvent shares the same property. To do this, let's define a set of probability density functions $p(v;k)$ such that

$$p(v) = \frac{1}{R}\sum_{k=1}^{R} p(v;k). \tag{48}$$

Substituting these $k$-dependent functions to $\sigma$ instead of $p(v)$, we emulate different orders of $v_k$ in (20a) and arrive to the spectral factor $\sigma$ explicitly depending on the rank $r$. Now, the equations for the characteristics of (46) take the form

$$\frac{\mathrm{d}\overline{\tau}}{\mathrm{d}r} = 0, \quad \frac{\mathrm{d}z}{\mathrm{d}r} = \frac{1}{N}\int \frac{vp(v;r)\mathrm{d}v}{1 - v\overline{\tau}}, \tag{49}$$

which leads to

$$z(r) = z_0 + \frac{1}{N}\int \frac{v\mathrm{d}v}{1 - v\overline{\tau}_0(z_0)}\int_0^r p(v;k)\mathrm{d}k = z_0 + \Sigma(\overline{\tau}_0(z_0);r), \tag{50}$$

and

$$\overline{\tau}(z,r) = \overline{\tau}_0(z - \Sigma(\overline{\tau};r)). \tag{51}$$

And, since the integral over $k$ in (50) behaves very much like the sum from (48), the self-energy $\Sigma(\overline{\tau};R)$ (and, hence, the quantity $\overline{\tau}(z,R)$) is indeed order-independent. Finally, since the mean resolvent $\overline{G}(z,r)$ from our equations also has the form $\overline{G}(z,r) = \overline{G}_0(z - \Sigma(\overline{\tau};r))$, its value for $r = R$ is order-independent too as it should be.

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
