# Peer review of "Emergent fractal phase in energy stratified random models"

_SciPost Physics, doi:SciPost Phys. 11, 101 (2021)_

## Round 1 · Referee Report · Anonymous (Referee 1) · 2021-9-3

Strengths

1- The study of the properties is very detailed and the results look reliable. 2- The model is new and instructive. 3- The paper is very clearly written.

Report

In this paper the authors study the effect of correlations in the off-diagonal terms on the spectral properties of long-range disordered random matrix models. In particular they consider a set of disordered random matrix models with translationally-invariant correlations in the hopping which interpolate continuously from the fully correlated Richardson's model to the Rosenzweig-Porter ensemble with translation-invariant correlations. Such set of models is constructed via an energy-stratified spectral structure of the hopping that allows one to gradually reduce the range of correlations.

In the completely correlated Richardson's limit there is measure zero of high-energy states that take the most spectral weight of the hopping and effectively screen the bulk states. Due to this effect, called cooperative shielding, the majority of states cannot be delocalized by the hopping terms, despite their long-range nature.

The main result of the paper is to convincingly show, both analytically and numerically, that (in contradiction with simple cooperative shielding arguments) as soon as the number of energy stratified states starts to grow with the system size, the bulk spectrum becomes delocalized but fractal.

Since all previously known analytical methods to tackle these kinds of problems are limited to the case of fully uncorrelated models, in order to calculate the fractal dimension of the bulk states the authors put forward a generalization of the cavity method and of the Dyson Brownian motion based on the spectral decomposition of the kinetic term and the Sherman-Morrison formula for the Green's function.

These methods yield the complete phase diagram of the model and demonstrate the existence of non-ergodic extended states as soon as one deviates from the completely correlated case. The authors also perform numerical simulations that confirm the validity of the analytical results.

The study of the properties is very detailed and the results look reliable. The model is new and interesting. Moreover the methods developed in the paper could be applied to a broader class of models. The paper is also very clearly written, very pedagogical, and pleasant to read.

Based on all this, I think that the paper is suitable for publication in SciPost.
  • validity: high
  • significance: high
  • originality: high
  • clarity: top
  • formatting: excellent
  • grammar: excellent

Author:  Ivan Khaymovich  on 2021-09-03  [id 1730]

(in reply to Report 1 on 2021-09-03)
Category:
remark

We are very grateful to the Referee for high evaluation of our work and acceptance of the manuscript as it is.

---

## Round 1 · Referee Report · Anonymous (Referee 2) · 2021-9-26

Strengths

  1. Useful technical results in a new family of models, interpolating between well known canonical models.
  2. New analytic technique presented.
  3. Writing is well paced, and with an explanatory style.

Weaknesses

  1. Results are not clearly organised/communicated, in particular no intuitive understanding of the results is given.

Report

In this paper the authors define a family of models which interpolate between Richardson’s Model (RM), and a translationally invariant form of the Rosenzweig-Porter model, (dubbed the TIRP).

They present analytic calculations of the fractal dimension using a technique which is developed in the manuscript, and provide some numerical evidence that shows agreement with these calculations. In particular authors explore how the principle of cooperative shielding, which is known to dictate the behaviour of the RM, breaks down as one tunes towards the TIRP.

The manuscript is well paced, and adopts a useful pedagogic style. However, despite this, it is unnecessarily difficult to follow, primarily due to (i) the lack of explicit definitions of the quantities of interest, and (ii) the lack of explanation of the physics underlying results.

However, my concerns about the manuscript are easily addressed, and overall I feel the results represent a useful addition to the literature. As a result recommend the manuscript for publication with corrections.

Requested changes

Major recommended change:

Organisation of results: My main concern with the paper is that the main results can be understood with simple physical intuitive arguments which are not clearly presented. I recommend that such short arguments should be included early in the paper, before delving into technical analysis, as they help guide the reader, and make the results accessible. In particular to readers who may be less interested in the details of the technical machinery which the authors develop.

I am sure the authors are aware of these simple arguments, but in the current version of the manuscript they are not highlighted in an accessible way.

Specifically, I refer to the fact the model has two regimes, which are distinguished by the hierarchy of the bandwidth of T ( = N^0) and the bandwidth of V ( = N^{1 - \gamma/2 - \beta / 2} ). In each of these regimes the fractal dimension can be found according to simple arguments (specifically: Energy stratification, and Fermi's golden rule).

  1. \beta + \gamma < 2: In this regime the bandwidth of V >> the bandwidth of T, and we consider the action of T on the diagonal orbitals of V. Due to its much smaller bandwidth T is unable to mix all of the diagonal orbitals of V, it is however able to mix the entropically dominant fraction of the diagonal orbitals of V, that is the N – N^\beta levels at zero energy. Naively one might expect that these degenerate orbitals would be localised by the potential T. However, (as the authors point out at the bottom of page 13) it is not possible to define a localised basis over the N – N^\beta remaining levels, as the N^\beta energy stratified levels are delocalised. The best one can do is define a basis in which the diagonal orbitals of H are spread over M = N^\beta of the diagonal orbitals of T. i.e. the fractal dimension of the diagonal orbitals of H (defined by M ~ N^D) is given by D = \beta. This obtains the phase diagram (Fig 4) to the left of the diagonal line.

  2. \beta + \gamma > 2: In this regime the fractal dimension may be calculated from Fermi’s golden rule (FGR). Specifically, as the bandwidth of V << bandwidth of T, and we may treat V as a perturbation which endows a finite lifetime to the diagonal orbitals of T. The FGR decay rate is the \Gamma = \rho_T |V_{ij} |^2 ~ N^{-1} |N^{-\gamma/2}|^2 = N^{1 - \gamma}. It follows that diagonal orbitals of T are superpositions of diagonal orbitals of H drawn from an energy window of width \Gamma ~ N^{1 - \gamma}, i.e. a number of orbitals M = \Gamma \rho_T = ~ N^{2 - \gamma}. By reciprocity it follows that the diagonal orbitals of H are superpositions of M ~ N^{2 - \gamma} diagonal orbitals of T, i.e. the diagonal orbitals have a fractal dimension D (defined by M ~ N^D) of D = 2 - \gamma. The only additional wrinkle is to note that M cannot exceed N or fall below 1, thus D = 1 if \gamma < 1 and D = 0 is \gamma > 2. This obtains the phase diagram (Fig 4) to the right of the diagonal line.

Minor changes:

  1. The authors should explicitly define the fractal dimension at least once, preferably early in the paper. There are many ways to define fractal dimensions, and while one may be considered a standard default definition in the analysis of wavefunctions defined on a lattice, the manuscript should be explicit.

  2. “However this guess is quite surprisingly fails …” -> “However, surprisingly, this guess fails …”

  3. “given by a simply looking formula” -> “given by the simple formula”

  4. Missing power of two in the denominator of the Lorentzian factor in Eq. 10

  5. I did not understand the footnote 9 (bottom of page 13). The authors are correct in their primary point: that an orthonormalised basis of orbitals defined over the N - N^\beta non-stratified levels typically consists of orbitals spread over at-least M = O(\beta) sites. However it is not true that a basis of N orthogonal orbitals projected onto N - N^\beta non-stratified levels is spread in the same way (i.e. it seems the orthonormality is an important ingredient). In particular it is easily checked that the plane waves given in the footnote are spread over O(1) sites in the limit of large N, specifically they have participation ratio PR = 1 + O(N^{\beta-1}).

  6. As a final comment, I am curious to ask, was it deliberate to use T to denote a potential in the localised basis, and V to denote a kinetic term, contravening the standard convention (T for kinetic energy, V for potential energy)?

  • validity: high
  • significance: good
  • originality: high
  • clarity: ok
  • formatting: excellent
  • grammar: good

Author:  Ivan Khaymovich  on 2021-10-29  [id 1889]

(in reply to Report 2 on 2021-09-26)
Category:
answer to question

For completness we put our detailed reply to the referee also here (not only in the resubmission comments):

Referee 2 report: Strengths 1. Useful technical results in a new family of models, interpolating between well known canonical models. 2. New analytic technique presented. 3. Writing is well paced, and with an explanatory style. Weaknesses 1. Results are not clearly organised/communicated, in particular no intuitive understanding of the results is given.

Reply: We thank the referee for the high evaluation of our work and for highlighting its results and hope that the revised version clearly represents our results, including the intuitive understanding of them.

Referee 2 report: The manuscript is well paced, and adopts a useful pedagogic style. However, despite this, it is unnecessarily difficult to follow, primarily due to (i) the lack of explicit definitions of the quantities of interest, and (ii) the lack of explanation of the physics underlying results.

Reply: In the revised version of the manuscript we have defined the quantity of interest explicitly and added the physical explanation of our results.

Referee 2 report: Major recommended change: Organisation of results: My main concern with the paper is that the main results can be understood with simple physical intuitive arguments which are not clearly presented. I recommend that such short arguments should be included early in the paper, before delving into technical analysis, as they help guide the reader, and make the results accessible. In particular to readers who may be less interested in the details of the technical machinery which the authors develop.

I am sure the authors are aware of these simple arguments, but in the current version of the manuscript they are not highlighted in an accessible way.

Specifically, I refer to the fact the model has two regimes, which are distinguished by the hierarchy of the bandwidth of T ( = N^0) and the bandwidth of V ( = N^{1 - \gamma/2 - \beta / 2} ). In each of these regimes the fractal dimension can be found according to simple arguments (specifically: Energy stratification, and Fermi's golden rule).

Reply: We agree with the point of the referee about intuitive explanation of our results, however, there are a couple of warnings which are worth to mention: 1) A blind usage of the Fermi’s golden rule for the system (including already the Richardson’s model) may lead to the wrong conclusion of the ergodicity in the entire region gamma<1. This is in some sense similar to the point based on [34, 35], when the divergence of the locator expansion series is associated to the ergodic delocalization of wave functions. The examples of [56-61] explicitly show that for the correlated hopping terms both these simple and physically intuitive approaches might fail.

2) As the follow-up of the previous argument, strictly speaking, for the correlated models one cannot use Fermi's golden rule even in the regime \beta+\gamma>2 of the considered models. Moreover, even the known methods like the cavity equations or Dyson Brownian motion are not able to overcome this problem. This forced us to develop our own method which takes into account all such correlations explicitly.

Of course, result-wise we completely agree with the Referee that the above simple intuitive arguments work for the considered range of models, but it cannot keep us away from possible errors in more involved correlated models.

---

## Round 2 · Author Response

Dear Editor,

We are grateful to both referees for the high evaluation of our paper and especially to referee 2 for thorough careful reading of the manuscript. His/her critical remarks have allowed us to significantly improve the presentation of our work.

In the revised version of our manuscript, we address all the points mentioned by the referee 2 (as the referee 1 has accepted our manuscript “as is”).

The point-to-point reply to the referee 2 is given below.

Sincerely yours, Anton G. Kutlin and Ivan M. Khaymovich.

Referee 2 report: Strengths 1. Useful technical results in a new family of models, interpolating between well known canonical models. 2. New analytic technique presented. 3. Writing is well paced, and with an explanatory style. Weaknesses 1. Results are not clearly organised/communicated, in particular no intuitive understanding of the results is given.

Reply: We thank the referee for the high evaluation of our work and for highlighting its results and hope that the revised version clearly represents our results, including the intuitive understanding of them.

Referee 2 report: The manuscript is well paced, and adopts a useful pedagogic style. However, despite this, it is unnecessarily difficult to follow, primarily due to (i) the lack of explicit definitions of the quantities of interest, and (ii) the lack of explanation of the physics underlying results.

Reply: In the revised version of the manuscript we have defined the quantity of interest explicitly and added the physical explanation of our results.

Referee 2 report: Major recommended change:

Organisation of results: My main concern with the paper is that the main results can be understood with simple physical intuitive arguments which are not clearly presented. I recommend that such short arguments should be included early in the paper, before delving into technical analysis, as they help guide the reader, and make the results accessible. In particular to readers who may be less interested in the details of the technical machinery which the authors develop.

I am sure the authors are aware of these simple arguments, but in the current version of the manuscript they are not highlighted in an accessible way.

Specifically, I refer to the fact the model has two regimes, which are distinguished by the hierarchy of the bandwidth of T ( = N^0) and the bandwidth of V ( = N^{1 - \gamma/2 - \beta / 2} ). In each of these regimes the fractal dimension can be found according to simple arguments (specifically: Energy stratification, and Fermi's golden rule).

Reply: We agree with the point of the referee about intuitive explanation of our results, however, there are a couple of warnings which are worth to mention: 1) A blind usage of the Fermi’s golden rule for the system (including already the Richardson’s model) may lead to the wrong conclusion of the ergodicity in the entire region gamma<1. This is in some sense similar to the point based on [34, 35], when the divergence of the locator expansion series is associated to the ergodic delocalization of wave functions. The examples of [56-61] explicitly show that for the correlated hopping terms both these simple and physically intuitive approaches might fail.

2) As the follow-up of the previous argument, strictly speaking, for the correlated models one cannot use Fermi's golden rule even in the regime \beta+\gamma>2 of the considered models. Moreover, even the known methods like the cavity equations or Dyson Brownian motion are not able to overcome this problem. This forced us to develop our own method which takes into account all such correlations explicitly.

Of course, result-wise we completely agree with the Referee that the above simple intuitive arguments work for the considered range of models, but it cannot keep us away from possible errors in more involved correlated models.

---

## Round 2 · List of Changes

The list of main changes (numbers are according to the minor changes suggested by the referee):

  1. (corresponds to the requested major changes) We added an explanation of the results to the end of Sec. 2 before their technical derivation. In addition, we have clarified the description of the model in the corresponding section in the pictorial way by adding a new figure.

  2. We defined the fractal dimension D explicitly via the inverse participation ratio (IPR) in Eq. (8) on p 6. In addition, in Eq. (13) of Sec. 3 we related the above fractal dimension to the local resolvent (Green’s function) which we calculate in the main part of the manuscript.

2., 3., 4. We corrected grammar and typos in the text and formulas.

  1. We removed footnote 9 and slightly rewrote the corresponding discussion. We thank the referee for pointing out that our example was incorrect: the orthogonality definitely matters, however this does not affect our general result. Besides, in the qualitative explanation of the results we added a footnote 4 clarifying the mechanism of how orthogonality leads to D(\beta)>=\beta.

  2. Finally, regarding your comment about using T for a potential and V for a kinetic term: it was not deliberate. We have changed the notation of the diagonal energy term T to H_0 in order to match the notations with the bunch of old works on the Rosenzweig-Porter and [16].

---

## Editorial Decision

published